# Recent Novel Hybrid Pd–Fe$_3$O$_4$ Nanoparticles as Catalysts for Various C–C Coupling Reactions

**Sanha Jang** [1],[†]**, Shamim Ahmed Hira** [1],[†] **, Dicky Annas** [1]**, Sehwan Song** [2]**, Mohammad Yusuf** [1]**, Ji Chan Park** [3]**, Sungkyun Park** [2] **and Kang Hyun Park** [1],[*]

[1]  Department of Chemistry, Pusan National University, Busandaehak-ro 63beon-gil, Geumjeong-gu, Busan 609-735, Korea

[2]  Department of Physics, Pusan National University, Busandaehak-ro 63beon-gil, Geumjeong-gu, Busan 609-735, Korea

[3]  Clean Fuel Laboratory, Korea Institute of Energy Research, Dajeon 305-343, Korea

[*]  Correspondence: chemistry@pusan.ac.kr; Tel.: +82-51-510-2238

[†]  These authors contributed equally to this work.

**Abstract:** The use of nanostructure materials as heterogeneous catalysts in the synthesis of organic compounds have been receiving more attention in the rapid developing area of nanotechnology. In this review, we mainly focused on our own work on the synthesis of hybrid palladium–iron oxide nanoparticles. We discuss the synthesis of Pd–Fe$_3$O$_4$—both morphology-controlled synthesis of Pd–Fe$_3$O$_4$ and transition metal-loaded Pd–Fe$_3$O$_4$—as well as its application in various C–C coupling reactions. In the case of rose-like Pd–Fe$_3$O$_4$ hybrid nanoparticles, thermal decomposition can be used instead of oxidants or reductants, and morphology can be easily controlled. We have developed a method for the synthesis of nanoparticles that is facile and eco-friendly. The catalyst was recyclable for up to five continual cycles without significant loss of catalytic activity and may provide a great platform as a catalyst for other organic reactions in the near future.

**Keywords:** hybrid catalyst; Pd–Fe$_3$O$_4$ nanoparticles; heterogeneous catalyst; C–C coupling reaction

## 1. Introduction

In recent years, fusion multimetallic nanoparticles (NPs) have generally been synthesized for use as catalysts, due to properties such as high selectivity for target material, catalytic activity, and physical/chemical stability when compared with equivalent catalysts based on a single metal [1–10]. In addition, their preparation has been optimized towards the design and synthesis via capping agent for controlled shape, size, and crystal structure [11–15]. In addition, various solvents—such as organic and inorganic solvents dispersed in hydrophilic or hydrophobic capping agents—are important for biological applications involving the effective dispersion of NPs [16,17].

In particular, homogeneous palladium catalysts have exhibited good performance with respect to reaction activities and turnover numbers (TONs). On the other hand, homogeneous catalysts have some decisive foibles, such as problems of recyclable and recovery, which lead to significant losses of costly metal [7]. Many studies have reported increased functionality using the incorporation of two or more clear nanomaterials [18–22]. Among various hybrid multimetallic NPs, palladium–iron oxide (Pd–Fe$_3$O$_4$) has attracted much attention owing to the high catalytic performance (Pd) and magnetically recoverable (Fe$_3$O$_4$) properties of each of the components of the nanocatalyst.

In a recent report, Hyeon et al. 2011 studied the facile synthesis of Pd–Fe$_3$O$_4$ NPs which were used to enable a catalytic effect for cross-coupling reactions. In addition, Wang and coworkers have reported Pd NPs embedded on carbon-coated Fe$_3$O$_4$ microspheres with magnetic properties. Chen et al. 2012 reported on

magnetically divisible hybrid Pd/Fe$_3$O$_4$@charcoal catalysts which are made up of active metal of 10 nm-sized Pd NPs and loaded in a 120 nm-sized iron oxide/carbon matrix [23–25].

As is well known, the Suzuki–Miyaura coupling, Mizoroki–Heck, and Sonogashira reactions using Pd catalyst—called C–C coupling reactions—are important in chemical, pharmaceutical, and agricultural industries [26–28]. Numerous previous works have reported the use of heterodimer Pd–Fe$_3$O$_4$ NPs applied in C–C coupling reactions [23], Pd/Fe$_3$O$_4$@C [29], and hyperbranched polyglycerol functionalized Pd/Fe$_3$O$_4$@SiO$_2$ catalyst [30].

In this review, we concentrate on latter exploitations in the synthesis of hybrid Pd–Fe$_3$O$_4$ nanocatalysts and various strategies for (1) urchin-like FePd–Fe$_3$O$_4$ for magnetic properties [31], (2) magnetically recoverable Pd–Fe$_3$O$_4$ hybrid nanocatalysts [32], (3) effectiveness of high metal-loaded NPs [7], (4) morphology impact of an organic capping agent of hybrid Fe$_3$O$_4$/Pd NPs [33], (5) rose-like Pd–Fe$_3$O$_4$ hybrid nanocomposites of morphology control via thermal decomposition, and [34,35] (6) various transition metal-loaded Pd–Fe$_3$O$_4$ heterobimetallic nanoparticles (Scheme 1) [36–39].

## C-C coupling reaction by novel hybrid Pd-Fe$_3$O$_4$ nanoparticles

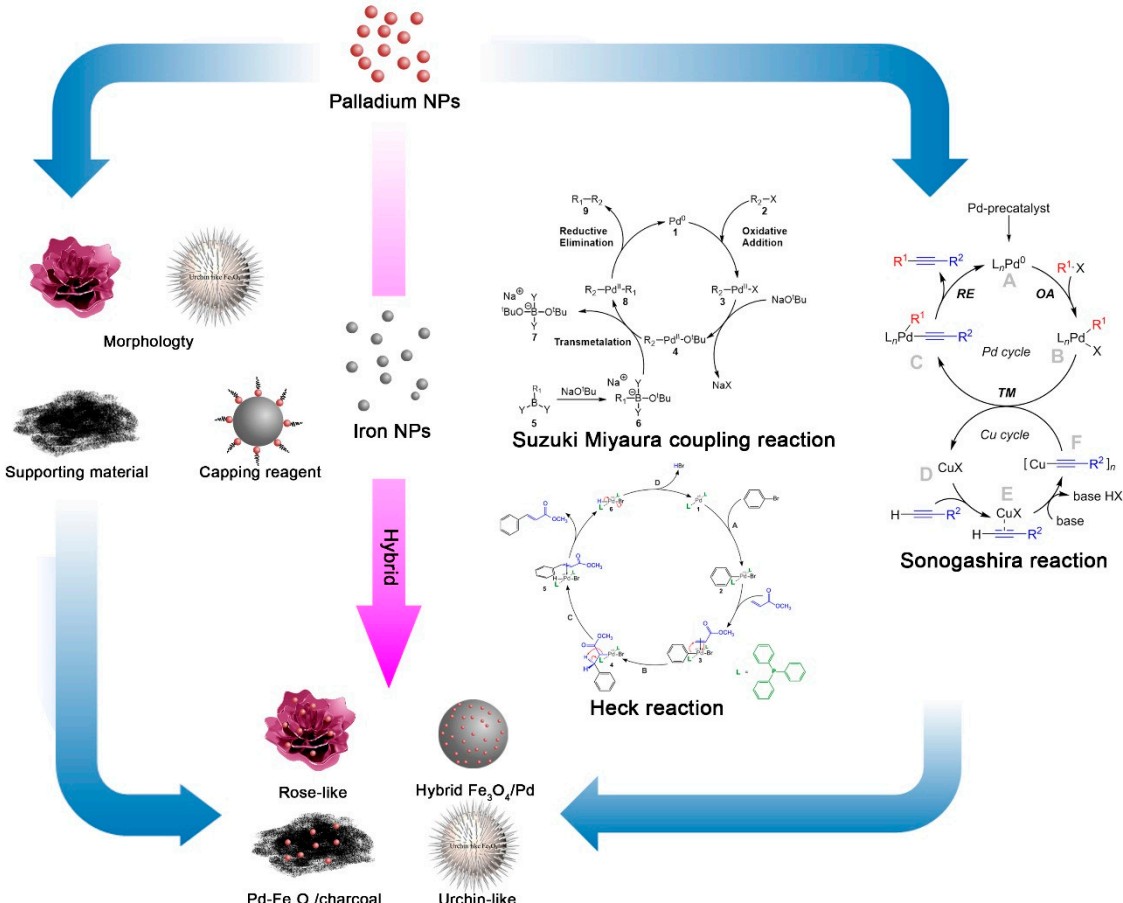

**Scheme 1.** C–C coupling reactions catalyzed by novel hybrid Pd–Fe$_3$O$_4$ nanoparticles.

## 2. Urchin-like FePd–Fe$_3$O$_4$: Nanocomposite Magnets

High saturation magnetization (M$_S$) and large magnetic coercivity (H$_C$) of magnetic materials are necessary for high-density power and data storage applications [40–42]. However, most magnetic materials contain only one of these two properties. For example, Fe, Co, and FeCo exhibit high M$_S$ and low H$_C$ (i.e., soft magnetic materials). By contrast, NdFeB and CoPt show low M$_S$ and high H$_C$ (i.e., hard magnetic materials). Thus, the exchange coupling between hard and soft materials,

with high Ms and Hc, has attracted much attention [43–46]. This concept was first proposed by F. Kneller and R. Hawing in 1991 [47], and there was a requirement that the size of the hard magnetic phase should be more than an almost half-domain wall width of the soft magnetic phase to maximize exchange coupling between the soft and hard magnet. However, experimentally, it is difficult to precisely fabricate the nanostructure strongly coupled with different magnetic properties into desirable nanosized materials [48,49]. FePt nanoparticles were firstly synthesized by Sun et al. [50] and have been demonstrated to be a suitable material for numerous magnetic nanocomposites due to huge magnetocrystalline anisotropy ($K_u \sim 6.6 \times 10^7$ erg cm$^{-3}$). After chemical reaction processing, FePt-based nanocomposites, such as FePt–Fe$_3$O$_4$ or FePt/Fe$_3$O$_4$ core–shell NPs, were converted to L1$_0$-FePt-based nanocomplexes that show high magnetic property [45,51]. However, sometimes, L1$_0$-FePt-Fe$_3$Pt complexes were formed that showed soft magnetic behavior due to thermodynamic instability. Therefore, it is difficult to synthesize strongly exchange-coupled nanocomposite magnets. Interestingly, FePd exhibits huge magnetocrystalline anisotropy ($K_u \sim 1.0 \times 10^7$ cm$^{-3}$). It also shows eutectoid reaction at exact temperatures depending on the Fe/Pd ratio. Therefore, it is possible to synthesize thermodynamically stable $\alpha$-Fe and L1$_0$-FePd through eutectoid reaction [52].

### 2.1. Synthesis of Urchin-Like Pd–Fe$_3$O$_4$ and L1$_0$-FePd–Fe Nanocomposite Magnets

Hayashi et al. reported the one-pot synthesis of urchin like FePd−Fe$_3$O$_4$ composites and their change into L1$_0$-FePd−Fe nanocomplex magnets [53]. Urchin-like nanocomposite with various Fe/Pd ratios (45:55, 49:51, 67:33, and 74:26) were synthesized in the following order. Figure 1 shows the synthetic process, illustration, and high-resolution TEM (HRTEM) image of urchin-like FePd–Fe$_3$O$_4$ composites (Scheme 2).

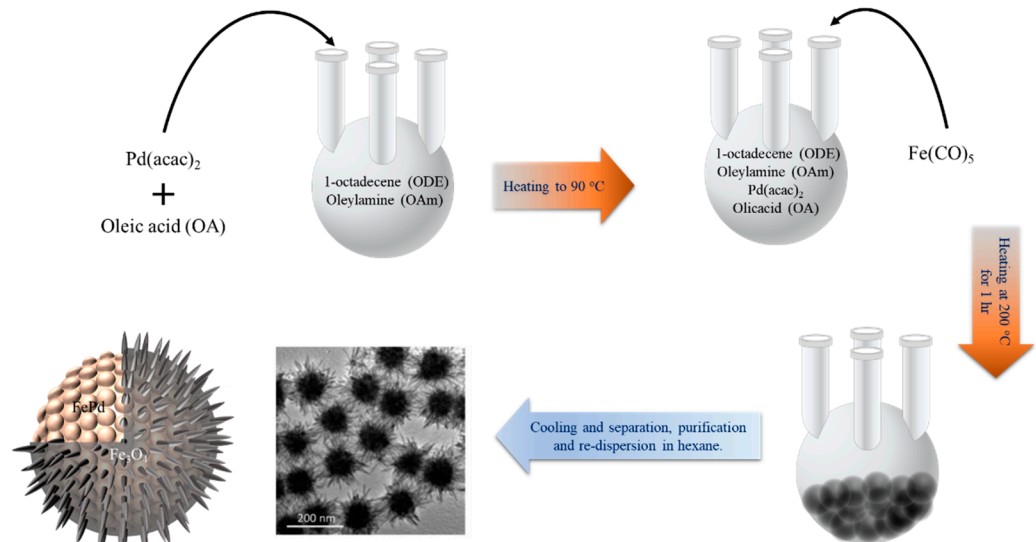

**Scheme 2.** Synthetic process, illustration, and TEM image of the urchin-like FePd–Fe$_3$O$_4$ composite nanoparticles.

These urchin-like FePd–Fe$_3$O$_4$ nanocomplexes were the precursors in the synthesis of Ll$_0$-FePd–Fe nanocomposite. Urchin-like FePd–Fe$_3$O$_4$ composites with various Fe/Pd ratios were annealed at various temperatures (350, 400, 450, 500, and 550 °C) under mixed gas conditions (Ar and H$_2$). These composites differed in phase according to annealing temperature and Fe/Pd ratio. Figure 1a shows a HRTEM image of L1$_0$-FePd–Fe nanocomposite magnet (Fe/Pd = 67:33) heat-treated at 500 °C that has two different domains. One of them is the (111) planes of L1$_0$-FePd with 0.27 nm lattice spacing. The other is (110) planes of $\alpha$-Fe with 0.20 nm lattice spacing. In addition, it is possible to distinguish L1$_0$-FePd from $\alpha$-Fe using energy dispersive spectroscopy (EDS) with elemental mappings of Pd (red) and Fe (blue). Pd (red) collaborating with Fe (blue) indicates L1$_0$-FePd, and separated Fe (blue) indicates $\alpha$-Fe (b–d).

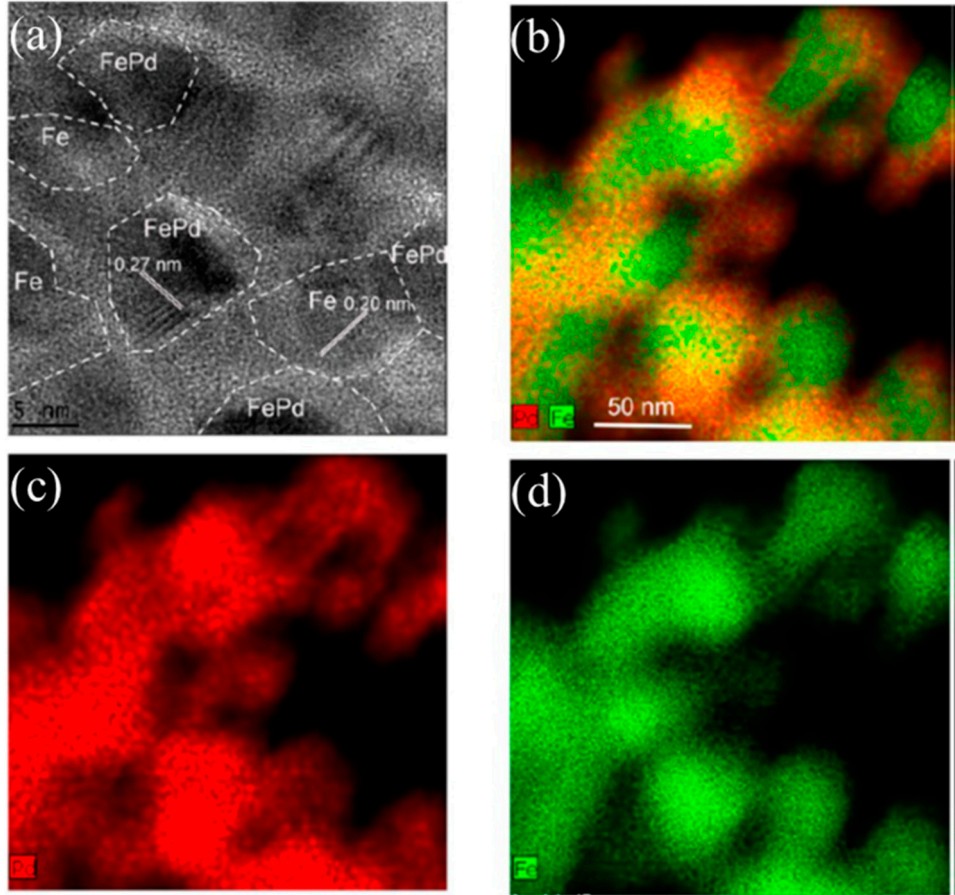

**Figure 1.** (**a**) HRTEM image of the L1$_0$-FePd–Fe nanocomposite grain with L1$_0$-FePd or Fe and (**b**) indicate EDS elemental mappings of Pd (red) and Fe (green) combined signals (**c**), (**d**) indicate single element Pd (red) (**c**) and Fe (green), respectively. Reproduced with permission from Sun, *Nano Letters*, published by American Chemical Society, 2013.

*2.2. Magnetic Properties of L1$_0$-FePd–Fe Nanocomposites Magnets*

Figure 2 summarizes the annealing temperature and Fe/Pd ratio dependent magnetic properties measured by vibrating sample magnetometer at room temperature (RT). Figure 2a shows magnetic hysteresis curves of Fe$_{67}$Pd$_{33}$–Fe$_3$O$_4$ nanocomposites annealed at 350 and 450 °C, respectively. Both hysteresis rings show a single-phase-like performance (no double hysteresis loop), indicating L1$_0$-FePd and Fe have exchange interaction. Figure 2b indicates heat-treated temperature reliant on M$_S$ and H$_C$ of Fe$_{67}$Pd$_{33}$–Fe$_3$O$_4$ nanocomposites. Both M$_S$ and H$_C$ increase with increasing annealing temperature over 500 °C, owing to the increases in L1$_0$-FePd phase and grain size of α-Fe. On the other hand, H$_C$ abruptly decreases at 550 °C due to materialization of the fcc FePd phase. Therefore, 500 °C is an optimum annealing temperature. Figure 2c shows the magnetic hysteresis loop of Fe$_{45}$Pd$_{55}$ and Fe$_{67}$Pd$_{33}$–Fe$_3$O$_4$ nanocomposite annealed at 500 °C. From the single phase-like hysteresis exchange coupling of L1$_0$-FePd and α-Fe is inferred. Figure 2d indicates the magnetic properties of L1$_0$-FePd–Fe nanocomplex magnets on the basis of Fe concentration, and it can be possible to tune exchange coupling among L1$_0$-FePd and α-Fe by controlling the ratio of Fe phase.

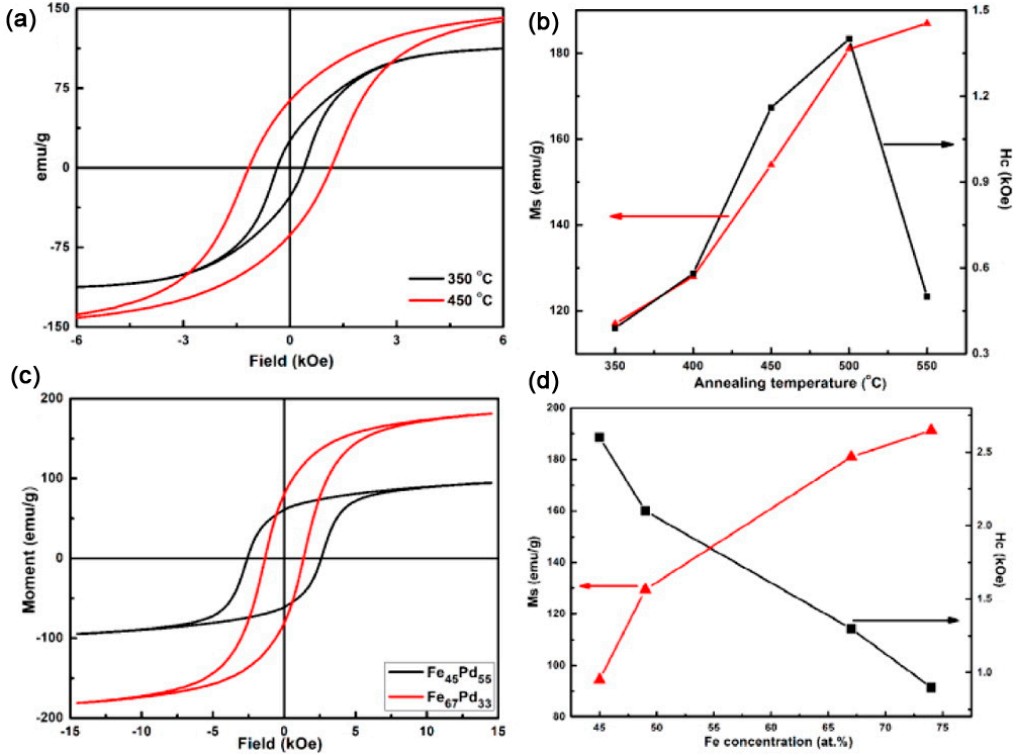

**Figure 2.** (**a**) Room temperature magnetic hysteresis loop of $Fe_{67}Pd_{33}$ heat-treated at 350 °C (black) and 450 °C (red), (**b**) indicates that annealing temperature is reliant on $M_s$ and $H_c$ of $Fe_{67}/Pd_{33}$–$Fe_3O_4$ nanocomposites. (**c**) Magnetic hysteresis loop of $Fe_{45}Pd_{55}$ (black) and $Fe_{67}Pd_{33}$–$Fe_3O_4$ (red) nanocomposite annealed at 500 °C, (**d**) Fe concentration reliant on Ms and $H_c$ for the complexes heat-treated at 500 °C. Reproduced with permission from Sun, *Nano Letters*, published by American Chemical Society, 2013.

## 3. Magnetically Recyclable Pd–Fe₃O₄ Hybrid Nanocatalyst: Application in Mizoroki–Heck Reaction

Pd is one of the most beneficial metal catalysts in organic synthesis for numerous C–C bond coupling reactions. On the other hand, $Fe_3O_4$ is one of the most used catalyst supports owing to its low price, easy separation, high magnetic properties, and easy reusability [54]. In organic synthesis, the Heck reaction is an important C–C coupling reaction which plays a significant role in medicinal, agrochemical, and fine chemical industries [55]. Li et al. [56] worked on one-step synthesis of Pd/Fe₃O₄ nanocomposites in 4-(2-hydroxyethyl)-1-piperazineethansulfonic acid (HEPES) buffer solution as an active catalyst for Suzuki coupling reaction. In another work, Li et al. [57] focused on the one-pot solvothermal synthesis of Pd/Fe₃O₄ nanocomposites as an environmental catalyst for Suzuki–Miyaura coupling reactions. Chung et al. [58] worked on Heck and Sonogashira coupling reactions using eco-friendly Pd–Fe₃O₄. Prasad et al. [59] worked on magnetically recyclable Pd–/Fe₃O₄-catalyzed Stille coupling reactions of organostannanes with aryl halides. Byun et al. [60] reported on systematic works of magnetically environmental Pd–Fe₃O₄ heterodimeric nanocrystal-catalyzed organic C–C coupling reactions. Elazab et al. [3] reported on highly efficient Pd–Fe₃O₄ on graphene support as a catalyst for Suzuki and Heck coupling reaction. Sreedhar et al. fabricated magnetically recyclable catalysts of Pd/Fe₃O₄ Hiyama coupling of aryl halides with aryl siloxanes [61]. The purpose of this work was to synthesis of Pd–Fe₃O₄ hybrid nanostructures and to evaluate their catalytic activity in Heck reaction.

### 3.1. Synthesis and Characterization of Pd–Fe₃O₄ Hybrid Nanocatalyst

The Pd–Fe₃O₄ was synthesized via a procedure modified from the literature (Scheme 3) [62,63].

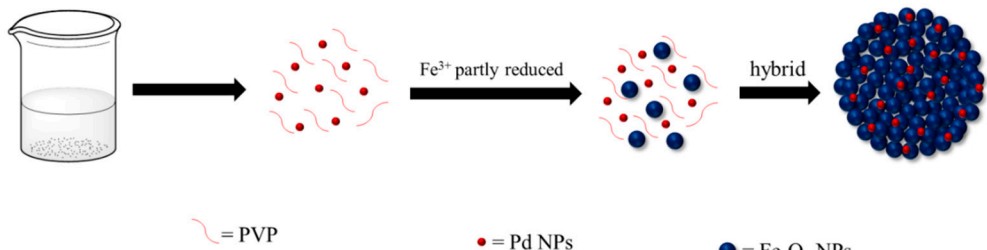

**Scheme 3.** Synthetic scheme of Pd–Fe$_3$O$_4$ hybrid nanocatalyst.

X-Ray Diffraction (XRD) was an effective tool to identify the existence of Pd nanoparticles on Fe$_3$O$_4$ (Figure 3). The diffraction peaks at 18.2, 30.1, 35.4, 43.1, 53.4, 56.9, and 62.9° (2 θ) correspond to (111), (220), (311), (400), (422), (511), and (440) planes of Fe$_3$O$_4$, and illustrate the fcc nature of Fe$_3$O$_4$ (JCPDS no. 19-0629). Similarly, the existence of peaks at 40.1, 46.5, and 68.0° (2 θ) are attributed to the (111), (200), and (220) plane of fcc Pd (JCPDS no. 46-1043).

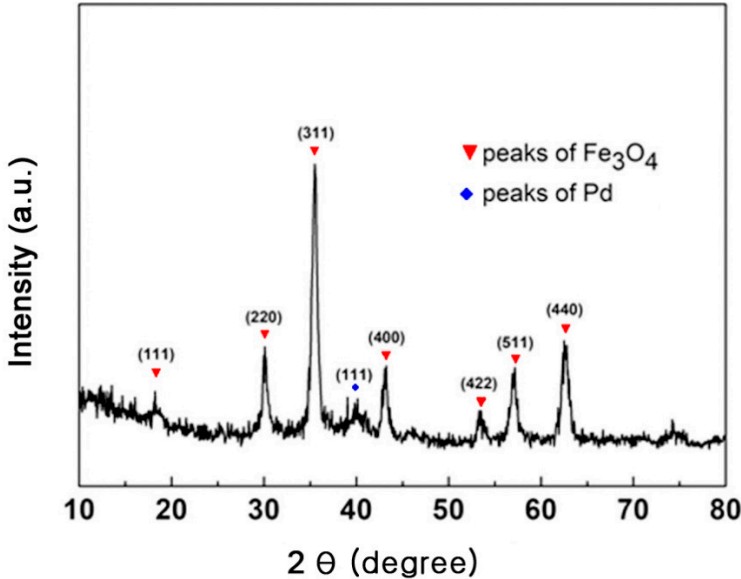

**Figure 3.** The XRD spectrum of Pd–Fe$_3$O$_4$.

The TEM images (Figure 4a,b) show spherically shaped nanoparticles with numerous cracks on the surface of the spheres, which suggest the porous structure of the sphere. The high-angle annular dark-field scanning TEM (HAADF-STEM) image (Figure 4c) and elemental mapping (Figure 4d,e) show that O and Fe are spread all over the composite structures, from the inside to the outside, illustrating a hybrid Pd–Fe$_3$O$_4$ arrangement. Figure 4f shows that most of the Pd NPs are scattered on the exterior of the hybrid nanocomplexes.

The value of saturation magnetization for the Pd–Fe$_3$O$_4$ was 58.8 emu g$^{-1}$. The supermagnetic characteristics of the Pd–Fe$_3$O$_4$ nanocomposites spheres are balanced with the fact that the spheres are compiled with major magnetic nanocrystals, which permits the nanoparticles to quickly gather together in the occurrence of an exterior magnetic field and effortlessly disperse in solution when the exterior magnetic field is eliminated, as shown in Figure 5b. The magnetization hysteresis curve of the synthesized Pd–Fe$_3$O$_4$ nanoparticles at 300 K is depicted in Figure 5c.

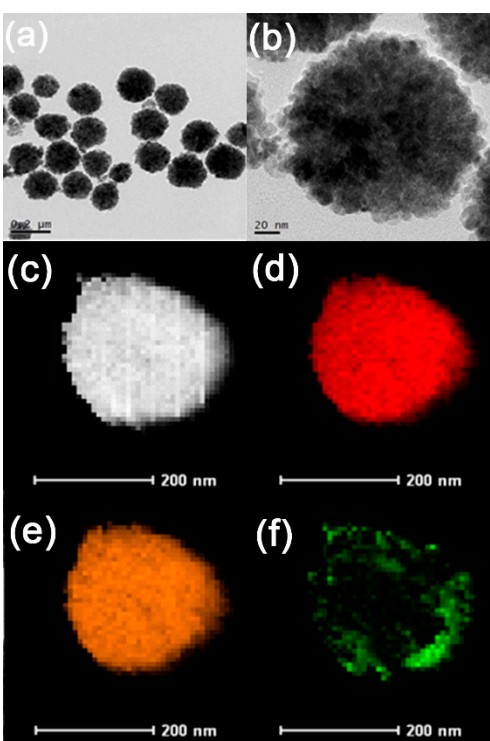

**Figure 4.** (**a**,**b**) TEM images of Pd–Fe$_3$O$_4$ at different magnifications, (**c**) HAADF-STEM image of Pd–Fe$_3$O$_4$, and EDX mapping of (**d**) O, (**e**) Fe, and (**f**) Pd.

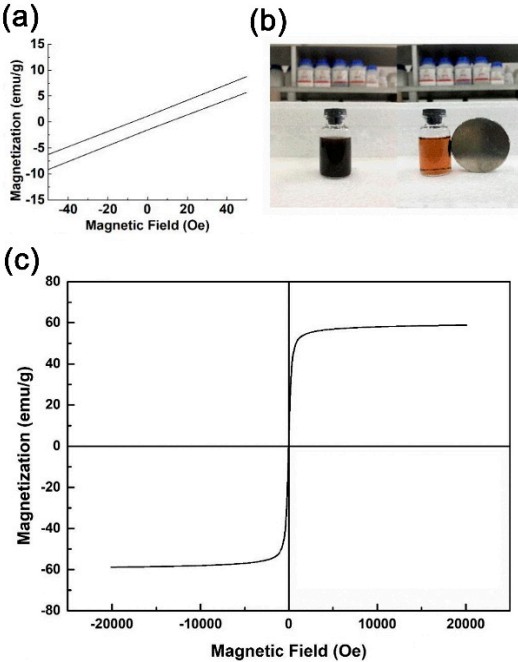

**Figure 5.** (**a**) The amplified magnetization hysteresis curves of Pd–Fe$_3$O$_4$; (**b**) The suspensions before and after magnetic separation by an external magnet; (**c**) The magnetization hysteresis loop of Pd–Fe$_3$O$_4$.

### 3.2. Pd–Fe$_3$O$_4$ Hybrid Nanocatalyst: Application in Mizoroki–Heck Reaction

Heck reaction of iodobenzene and styrene was selected as a model reaction to investigate the catalytic efficiency and stability of the hybrid nanostructures (Table 1). Firstly, the impact of catalyst loading was observed by using various amounts of catalysts in the range of 0.054 to 0.162 mol % (entries 1–3, Table 1). The best performance was obtained with 0.108 mol % (entry 2, Table 1). Then, the effect

of temperature was also analyzed. Increasing the reaction temperature from 60 to 110 °C had a crucial effect on the advancement of the reaction (entries 2, 4–7, 9, and 10, Table 1). It can be noted that even at low temperature (80 °C), the reaction still gave 51.3% yield in 3 h (entry 6, Table 1). With increases in reaction time, the yield also increased (entry 8, Table 1).

**Table 1.** The reaction of iodobenzene and styrene catalyzed by Pd–Fe$_3$O$_4$ [a].

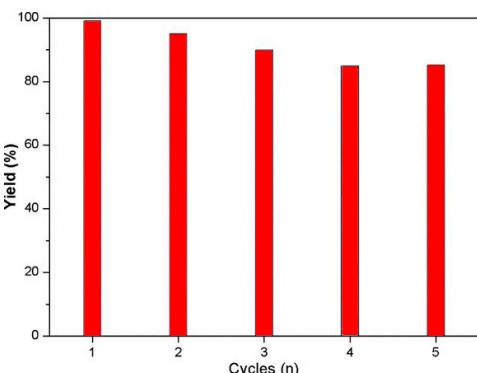

| Entry | Pd added [b] | Temp (°C) | Time (h) | Yield [c] (%) |
|-------|--------------|-----------|----------|---------------|
| 1 | 0.054 | 110 | 3 | 93 |
| 2 | 0.108 | 110 | 3 | 99 |
| 3 | 0.162 | 110 | 3 | 99 |
| 4 | 0.108 | 60 | 3 | 19.8 |
| 5 | 0.108 | 70 | 3 | 34.5 |
| 6 | 0.108 | 80 | 3 | 51.3 |
| 7 | 0.108 | 80 | 6 | 57.1 |
| 8 | 0.108 | 80 | 24 | 91.2 |
| 9 | 0.108 | 90 | 3 | 69.2 |
| 10 | 0.108 | 100 | 3 | 85.4 |

[a] Reaction conditions: iodobenzene (5 mmol), styrene (7.5 mmol), NEt$_3$ (7.5 mmol), DMF (10 mL) and nitrogen atmosphere. [b] Relative to the amount of iodobenzene. [c] The products were investigated by GC using an internal standard (decane).

Durability is an essential factor to analyze the practical applicability of a catalyst. The synthesized catalyst was utilized for the Heck coupling reaction of iodobenzene with styrene at 110 °C for 3 h. After finishing the reaction, it was possible to separate the catalyst using an exterior magnet. It was washed with EtOH and H$_2$O and dried under the vacuum. The catalyst was then used for the next catalytic cycle. After five runs, the yield of the targeted product decreased from 99% to 85% (Figure 6).

**Figure 6.** The recyclability test of Pd–Fe$_3$O$_4$ for Heck reaction over five consecutive cycles.

The TEM image of reused Pd–Fe$_3$O$_4$ nanoparticles showed that they kept their sphere-shaped structure after being used for five catalytic cycles (Figure 7a). To recognize the actual catalytic active sites through the reaction, a hot-filtration experiment was necessary (Figure 7b). A Heck reaction was carried out by taking a similar substrate was that the catalytic activity was filtered off after reacting for 0.5 h (55% yield). Using the same reaction conditions, the resulting solution was heated, and the reaction was immobile after 1 h (75% yield). This indicated that the leached Pd species can also result in activation of the catalyst for Heck reaction. On the other hand, owing to the lack of Pd (supported Pd nanoparticles), the reaction could not continue. As a result, for Pd–Fe$_3$O$_4$ hybrid nanocatalysts, both Pd leaching into the reaction complex and the supported Pd nanoparticles are the vigorous catalytic species, and the Heck coupling reaction begins partially homogenously in the complex, even if the initial catalyst is heterogeneous.



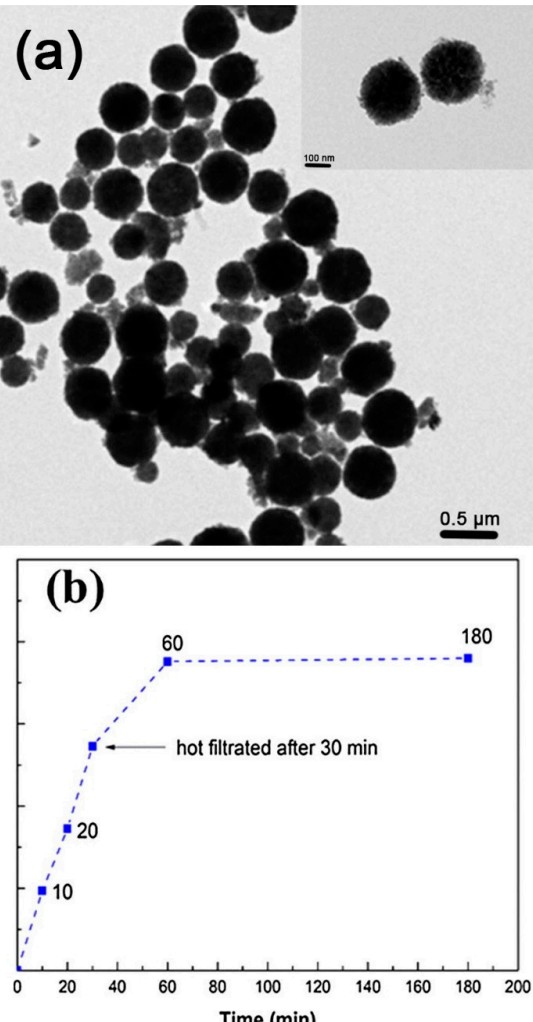

**Figure 7.** (**a**) TEM image of recycled Pd–Fe$_3$O$_4$ after five cycles and (**b**) hot-filtration experiment in the Heck reaction of iodobenzene and styrene.

Put simply, a highly active Pd–Fe$_3$O$_4$ has been synthesized using a facile method. The Pd–Fe$_3$O$_4$ showed good magnetic properties and exhibited good dispersion in solvent. At low temperature (80 °C), the catalyst exhibited excellent catalytic activity. This suggests a number of benefits, including easy synthesis, excellent reactivity, and perfect robustness. The magnetic characteristics of the catalyst permit it to be segregated and recoverable.

## 4. Bifunctional Catalyst of Pd/Fe$_3$O$_4$/C: High content of Nanoparticles

High metal loading and high dispersion greatly influences the catalytic activity of nanoparticles due to the use of material with a high pore volume [7,64]. Some materials, such as silica or carbon, have been used as supporting materials due to their high pore volume, so that many metals can be loaded in the pore [65,66]. High pore volume materials have been used to avoid agglomeration of nanoparticles [67]. Carbon materials have been developed as supporting materials for various metals with good thermal stability and mechanical stability [21]. Activated charcoal can be used as a supporting material because charcoal has a low price, large pore volume, and large surface area [7].

Due to some advantages, the use of charcoal can result in the high metal loading of nanoparticles, such as Pd and Fe$_3$O$_4$ [7]. Many researchers have used Pd and Fe$_3$O$_4$ nanoparticles as catalysts in various synthesis organic chemistry reactions, such as Suzuki–Miyaura coupling reaction [7,64]. Besides using supporting material, the stability of Pd nanoparticles can be increased by immobilizing Pd

nanoparticles on magnetic nanoparticles [23,68,69]. Magnetic nanoparticles, such as $Fe_3O_4$, have unique physical properties and possess several advantages, such as easy preparation, easy separation, low toxicity, and low cost [70–72]. $Fe_3O_4$ nanoparticles also have good catalytic activity because of the presence of two metals in the supporting material that increase catalytic activity in organic reactions [25].

### 4.1. Melt Infiltration Method

#### 4.1.1. Synthesis of Pd/Fe₃O₄/Charcoal Catalyst and Suzuki–Miyaura Coupling Reaction

High metal loading and uniformity of particle size in supported material are very important to increase the catalytic activity of the material. Magnetic Pd/$Fe_3O_4$/charcoal catalyst was successfully synthesized using various methods. One of them involved using mixed metal hydrate salt via the solid-state grinding method and without the addition of surfactant [7] (Scheme 4).

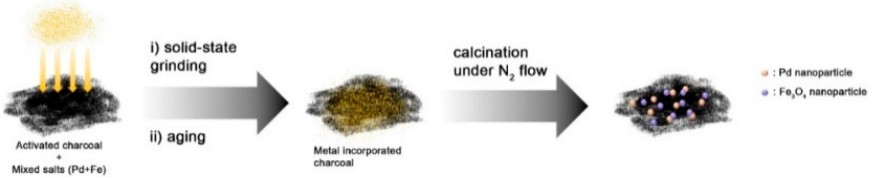

**Scheme 4.** The synthetic process of Pd/$Fe_3O_4$/charcoal nanoparticles.

Pd/$Fe_3O_4$/charcoal catalyst was successfully synthesized using two steps. The first was co-solid milling and the second was thermal decomposition under $N_2$ gas flow (Scheme 4). $Pd(NO_3)_3 \cdot 2H_2O$ and $Fe(NO_3)_3 \cdot 9H_2O$ were used as hydrate salts for the synthesis of Pd/$Fe_3O_4$ nanoparticles. In the co-solid grinding method, both hydrate salts were melt-infiltrated and entered into charcoal pores. Furthermore, thermal decomposition was carried out in 400 °C under $N_2$ conditions to form Pd and $Fe_3O_4$ nanoparticles.

Pd and $Fe_3O_4$ were successfully located in the charcoal using TEM imaging (Figure 8a). Based on HRTEM analysis in Figure 8b, Pd has been loaded with an average diameter size of 5 nm with a lattice distance of 0.255 nm ((111) planes of Pd). In addition, $Fe_3O_4$ with a greater size than Pd was also observed with the diameter around 9 nm and lattice distance of 0.253 nm ((311) planes). XRD analysis result shows that the Pd on Pd/$Fe_3O_4$/charcoal has a fcc structure. In this study, Pd hydrate salt was used at 20 wt %. Based on the ICP–AES result, the amount of Pd in the composition was 19.2 wt %. This shows that this method produces high metal loading on porous charcoal.

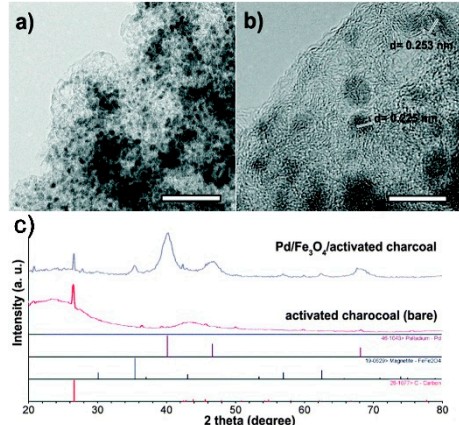

**Figure 8.** (**a**) TEM image with bar scale of 50 nm, (**b**) HRTEM image with bar scale of 5 nm, and (**c**) XRD spectra Pd/$Fe_3O_4$/charcoal, respectively. Reproduced with permission from Park, *New J. Chem.*, published by The Royal Society of Chemistry, 2014.

In Table 2, Pd/Fe$_3$O$_4$/charcoal was applied as heterogeneous catalyst in the Suzuki–Miyaura coupling reaction. 4-Bromoanisole and phenylboronic acid were reacted with in the presence of potassium carbonate. DMF/water (4:1) is the best solvent for this reaction with a turnover frequency of 25. The best result was obtained by adding Pd (20 wt %)/Fe$_3$O$_4$(10 wt %)/charcoal in the reaction with %conversion of >99%. This result is better compared to commercial Pd/charcoal catalyst. The high conversion of reactant into product using Pd (20 wt %)/Fe$_3$O$_4$(10 wt %)/charcoal catalyst is due to the uniform and small size of particles.

**Table 2.** Screening of optimum conditions in Suzuki–Miyaura coupling reactions. Reproduced with permission from Park, *New J. Chem.*, published by The Royal Society of Chemistry, 2014.

| Entry | Catalysts | Temp (°C) | Time (h) | Solvent | Conv. [a] (%) | Product Time Yield ($g_{product}\ g_{Pd}^{-1}\ h^{-1}$) |
|---|---|---|---|---|---|---|
| 1 | Pd/Fe$_3$O$_4$/charcoal | 150 | 30 | Toluene/H$_2$O (4:1) | 59 | 3.62 |
| 2 | Pd/Fe$_3$O$_4$/charcoal | 100 | 4 | Toluene/H$_2$O (4:1) | 4 | 1.84 |
| 3 | Pd/Fe$_3$O$_4$/charcoal | 100 | 4 | DMSO/H$_2$O (4:1) | 25 | 12.0 |
| 4 | Pd/Fe$_3$O$_4$/charcoal | 100 | 4 | THF/H$_2$O (4:1) | 35 | 15.1 |
| 5 | Pd/Fe$_3$O$_4$/charcoal (Pd 0.5 mol %) | 100 | 4 | DMF/H$_2$O (4:1) | 60 | 55.2 |
| 6 | Pd/Fe$_3$O$_4$/charcoal | 100 | 2 | DMF/H$_2$O (4:1) | 60 | 55.2 |
| 7 | Pd/charcoal [b] | 100 | 4 | DMF/H$_2$O (4:1) | 76 | 35.0 |
| 8 | Pd/Fe$_3$O$_4$/charcoal | 100 | 4 | DMF/H$_2$O (4:1) | >99 | 36.0 |
| 9 | Fe$_3$O$_4$/charcoal | 100 | 4 | DMF/H$_2$O (4:1) | No reaction | |
| 10 | Commercial Pd/charcoal | 100 | 4 | DMF/H$_2$O (4:1) | 31 | 14.3 |

The reactions were conducted using 20 wt % of Pd/Fe$_3$O$_4$/charcoal catalyst, 1 mmol of 4-bromoanisole, 1.2 mmol of phenylboronic acid, 2 mmol of K$_2$CO$_3$, 10 mL of DMF, and 2.5 mL of H$_2$O. [a] Analyzed using $^1$H NMR and [b] 20 wt% Pd NPs were used.

Table 3 summarized of scope of substrate in Suzuki–Miyaura coupling reaction using Pd/Fe$_3$O$_4$/charcoal catalyst. This catalyst can catalyze the reactions with good to excellent yield. This showed that Pd/Fe$_3$O$_4$/charcoal catalyst has good activity in this reaction. High metal loading, particle size, and uniformity of particles were the factors affected this reaction and producing a high conversion.

**Table 3.** Suzuki–Miyaura coupling reactions of aryl halides with arylboronic acid. Reproduced with permission from Park, *New J. Chem.*; published by The Royal Society of Chemistry, 2014.

| Entry | Aryl Halide | Arylboronic Acid | Product | Yield [a] (%) | Product Time Yield ($g_{product}\ g_{Pd}^{-1}\ h^{-1}$) |
|---|---|---|---|---|---|
| 1 | bromobenzene | (HO)$_2$B–phenyl | biphenyl | >99 | 36.2 |
| 2 | chlorobenzene | (HO)$_2$B–phenyl | biphenyl | >99 | 36.3 |
| 3 | phenyl–OTf | (HO)$_2$B–phenyl | biphenyl | >99 | 36.2 |
| 4 | 1-bromo-4-fluorobenzene | (HO)$_2$B–phenyl | F–biphenyl | >99 | 43.1 |
| 5 | 4-bromoanisole (OMe) | (HO)$_2$B–C$_6$H$_4$–CH$_3$ | MeO–biphenyl–CH$_3$ | 45 | 20.9 |
| 6 | 4-bromoanisole (OMe) | (HO)$_2$B–C$_6$H$_4$–CH$_3$ | MeO–biphenyl–CH$_3$ | 64 | 37.9 |
| 7 | 2-bromotoluene (CH$_3$) | (HO)$_2$B–phenyl | 2-methylbiphenyl (CH$_3$) | 48 | 18.9 |

**Table 3.** *Cont.*

| Entry | Aryl Halide | Arylboronic Acid | Product | Yield [a] (%) | Product Time Yield ($g_{product} \, g_{Pd}^{-1} \, h^{-1}$) |
|---|---|---|---|---|---|
| 8 | | $(HO)_2B\text{—}$ | | 99 | 48.6 |
| 9 | | $(HO)_2B\text{—}$ | | 83 | 35.5 |
| 10 | | $(HO)_2B\text{—}$ | | 90 | 38.5 |

The reactions were conducted using 20 wt % of Pd/Fe$_3$O$_4$/charcoal catalyst, 1 mmol of 4-bromoanisole, 1.2 mmol of phenylboronic acid, 2 mmol of K$_2$CO$_3$, 10 mL of DMF, and 2.5 mL of H$_2$O. [a] Analyzed using $^1$H NMR.

### 4.1.2. Recycling and Pd Leaching Test

Pd(20 wt %)/Fe$_3$O$_4$(10 wt %)/charcoal can be separated using an external magnet upon completion of the reaction, due to the superparamagnetic character of Fe$_3$O$_4$ which makes it possible to reuse several times. After being used three times (Table 4), it produced high catalysis activity with a conversion of >99%. The HRTEM images of the recovered Pd/Fe$_3$O$_4$/charcoal has been shown in Figure 9.

**Table 4.** Recycling test of Pd/Fe$_3$O$_4$/charcoal catalyst. Reproduced with permission from Park, *New J. Chem.*; published by The Royal Society of Chemistry, 2014.

| Recycle Run | Temp (°C) | Time (h) | Solvent | Conv. [a] (%) | Product Yield Time ($g_{product} \, g_{Pd}^{-1} \, h^{-1}$) |
|---|---|---|---|---|---|
| 1 | 100 | 4 | DMF/H$_2$O (4:1) | >99 | 46.0 |
| 2 | 100 | 4 | DMF/H$_2$O (4:1) | >99 | 46.0 |
| 3 | 100 | 4 | DMF/H$_2$O (4:1) | >99 | 46.0 |

The reactions were conducted using Pd/Fe$_3$O$_4$/charcoal catalyst with optimum conditions in Table 2 (entry 8). [a] Analyzed using $^1$H NMR.

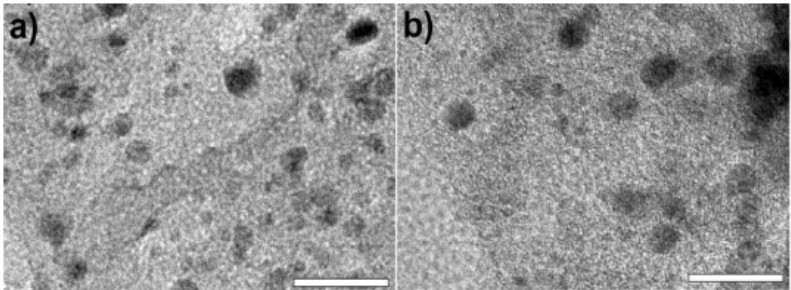

**Figure 9.** HRTEM images of the recovered Pd (20 wt %)/Fe$_3$O$_4$(10 wt %)/charcoal catalysts after recycling (**a**) two times and (**b**) three times. All bars represent 20 nm. Reproduced with permission from Park, *New J. Chem.*; published by The Royal Society of Chemistry, 2014.

After Suzuki–Miyaura coupling reaction, the filtrate solution was characterized using ICP–AES. The result showed that the Pd level was 0.48 ppm, which is negligible, and that there was almost no Pd leaching at the catalyst during the Suzuki–Miyaura coupling reaction.

### 4.2. Stöber Method

#### 4.2.1. Synthesis of Fe₃O₄@C–Pd Catalyst and Suzuki–Miyaura Coupling Reaction

Traditionally, the synthesis of silica spheres uses the Stöber method. According to a recent report, some studies successfully applied an improved Stöber method in the synthesis of resin spheres composed of resorcinol–formaldehyde (RF) that were transformed to carbon spheres (Scheme 5) [73–76].

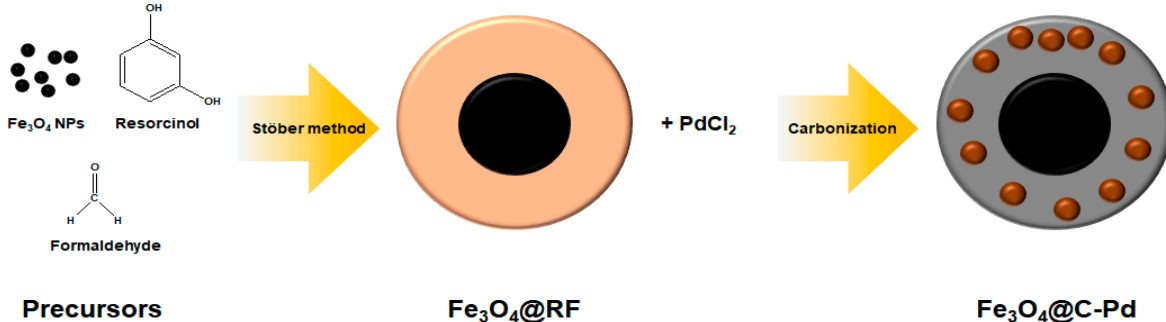

**Scheme 5.** The synthetic process of $Fe_3O_4$@C–Pd nanoparticles.

Pd was effectively embedded in the silica based on TEM imaging (Figure 10a). In the HRTEM analysis in Figure 10b, Pd has been loaded with an average diameter size of 10 nm with an interplanar spacing of 0.223 nm ((111) planes of Pd). In this study, the Pd content in the catalysts were measured by ICP–AES and the Pd loading amount reached 8.73 wt %.

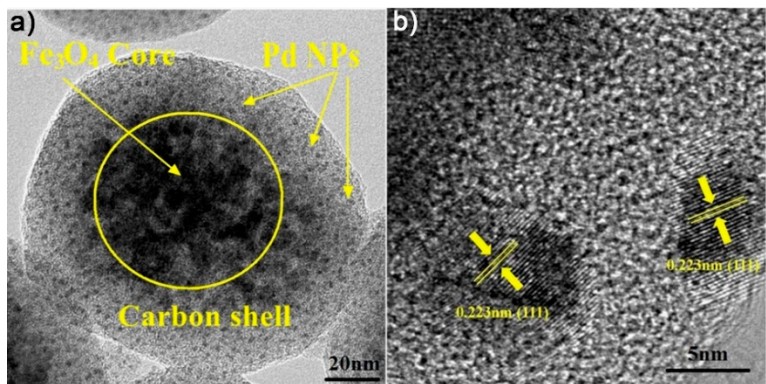

**Figure 10.** TEM image of (**a**) $Fe_3O_4$@C–Pd-350 and (**b**) HRTEM image of $Fe_3O_4$@C–Pd-550, respectively.

#### 4.2.2. Catalytic Efficiency of Fe₃O₄@C–Pd-550 Nanocomposite

$Fe_3O_4$@C–Pd-550 nanocomposite was used as a catalyst for producing structurally diverse aryl halides (Table 5). The reaction gave good yield in the case of aryl bromide compared with aryl chloride (entries 1 and 2). On the other hand, the aryl bromide containing electron-withdrawing group gave better yield than electron-donating group (entries 3 and 4).

**Table 5.** The Suzuki coupling reactions of structurally different aryl halides.

| Entry | A | Halogen | Time (h) | Yield (%) |
|-------|------|---------|----------|-----------|
| 1 | H | Br | 6 | 92.22 |
| 2 | H | Cl | 12 | 63.74 |
| 3 | $NO_2$ | Br | 6 | 94.51 |
| 4 | $OCH_3$ | Br | 6 | 87.12 |

## 5. Hybrid Fe$_3$O$_4$/Pd Catalysts: Impact of Organic Capping Agents

Fe$_3$O$_4$ nanoparticle is a heterogeneous catalyst that is easy to use and can be used repeatedly, which makes it an environmentally friendly catalyst [77]. The synthesis of Fe$_3$O$_4$ nanoparticles can be carried out in various ways, such as using co-precipitation [78]; electrochemical [79], sonochemical [80], and microemulsion techniques [81]; and hydrothermal processes [82]. Fe$_3$O$_4$ nanoparticles can be supported by adding other materials to the surface of Fe$_3$O$_4$ to provide support as a heterogeneous catalyst system, such as by adding Pd nanoparticles, and this is desired due to the ability of Pd as a catalyst with high reaction speed and high turnover rate (TON) [33].

Pd/Fe$_3$O$_4$ microsphere nanoparticles have attracted the attention of researchers because of their good catalytic activity [7]. This is because of the stability of the Pd/Fe$_3$O$_4$ microsphere when dispersed in organic and inorganic solvents. The dispersion stability can be controlled using hydrophilic or hydrophobic capping agents and, also, surface modification [16,17]. Pd/Fe$_3$O$_4$ microspheres were widely developed due to their advantage of being easily separated from the product using external magnets [7]. Several techniques have been advanced for the synthesis of Pd/Fe$_3$O$_4$ microspheres using a variety of capping agents to enhance the dispersion stability of Pd/Fe$_3$O$_4$ nanoparticles in solvents. Some capping agents have been used to improve dispersion stability, such as chitosan [83–85], metal [86], SiO$_2$ [87], and carbon [88], among others.

### 5.1. Immobilization of Pd NPs onto Each Fe$_3$O$_4$ Microsphere

The high dispersion stability of Pd/Fe$_3$O$_4$ nanoparticles using various capping agents can increase catalytic activity in organic reactions. Functional groups, such as poly(vinylpyrrolidone) (PVP), sodium citrate (Na$_3$Cit), and poly(ethylene glycol) (PEG) can be used as capping agents for the synthesis of Fe$_3$O$_4$ using a solvothermal method, and these have different levels of dispersion in water [33] (Scheme 6). Fe$_3$O$_4$ has microsphere shapes with different sizes. Fe$_3$O$_4$ microspheres with Na$_3$Cit as capping agent have a smaller average size than Fe$_3$O$_4$ with PEG and Fe$_3$O$_4$ with PVP. Fe$_3$O$_4$ synthesized with capping agents have smaller average size compared to Fe$_3$O$_4$ synthesized without capping agent.

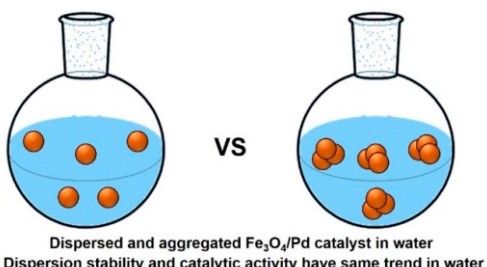

Dispersed and aggregated Fe$_3$O$_4$/Pd catalyst in water
Dispersion stability and catalytic activity have same trend in water

**Scheme 6.** Dispersion stability of Pd/Fe$_3$O$_4$ nanoparticles using capping agent in water.

Pd nanoparticles were immobilized to the Fe$_3$O$_4$ microsphere. Based on TEM images, Fe$_3$O$_4$ microsphere particles have average size of 3 nm. Meanwhile, the average size of Pd nanoparticles immobilized onto Fe$_3$O$_4$ microsphere with various capping agents in an aqueous solution was 4 nm, and they showed good dispersion with the exception of PVP–Fe$_3$O$_4$ microspheres (Figure 11, Table 6).

**Table 6.** Size distribution of Pd/Fe$_3$O$_4$ microspheres using various capping agents.

| Recycle Run | Capping Agent | Average Size (nm) |
|:-----------:|:-------------:|:-----------------:|
| 1 | Na$_3$Cit | 3.3 ± 0.24 |
| 2 | PEG | 3.4 ± 0.21 |
| 3 | PVP | 4.1 ± 0.43 |
| 4 | No capping agent | 4.7 ± 0.35 |

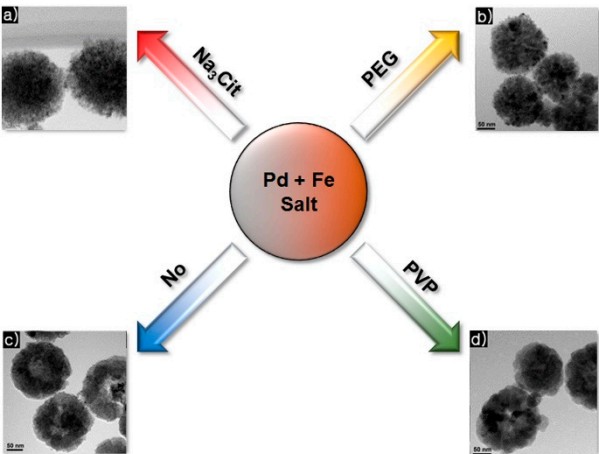

**Figure 11.** TEM images of immobilized of Pd onto $Fe_3O_4$ using various capping agents.

Based on XRD in Figure 12, $Fe_3O_4$ microsphere exhibited fcc structure with (111), (200), and (220) reflection (JCPDS No. 46-1043). In addition, there was a cubic spinel structure with (220), (311), (400), (422), and (511) reflection (JCPDS No. 19-0629).

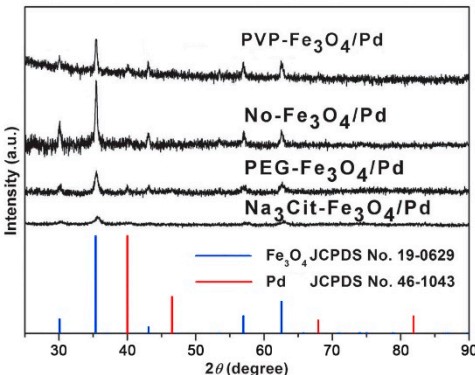

**Figure 12.** XRD spectrum of each $Fe_3O_4$/Pd catalysts. Reproduced with permission from Park, *ChemCatChem*; published by Wiley, 2014.

### 5.2. Suzuki Coupling Reaction Using Pd/Fe₃O₄ Nanoparticles with Various Capping Agents

Pd/$Fe_3O_4$ microsphere was used in Suzuki coupling reaction as a catalyst. Table 7 showed the effects of catalysts, temperatures, reaction times, solvents, and bases on the Suzuki coupling reaction using phenylboronic acid and bromobenzene. The optimum conditions were obtained when reacting phenylboronic acid and bromobenzene using $Na_3Cit$–$Fe_3O_4$/Pd (0.05 mol %), $H_2O$, 50 °C, and 7 h reaction time with %yield of 98% (entry 5). In addition, raising the temperature to 100 °C and reducing the reaction time to 1.5 h resulted in a high %yield (98%, entry 11). When comparing between Pd/$Fe_3O_4$ with $Na_3Cit$ used as a capping agent and Pd/$Fe_3O_4$ using other capping agents, $Na_3Cit$–$Fe_3O_4$/Pd had better effectiveness in producing high yield

**Table 7.** [a] Screening of optimum conditions in Suzuki coupling reactions. Reproduced with permission from Park, *ChemCatChem.*, published by Wiley, 2014.

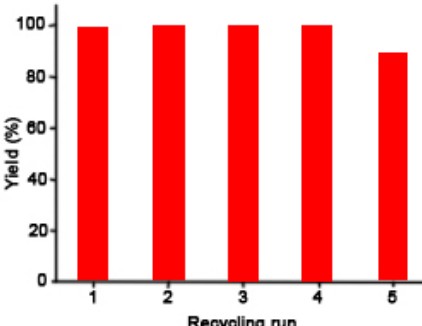

| Entry | Cat. (mol %) | Temp (°C) | Time (h) | Base | Solvent | Yield [b] (%) |
|---|---|---|---|---|---|---|
| 1 | 1 (Na₃Cit) | 80 | 5 | $K_2CO_3$ | $DMF/H_2O$ (4:1) | 91 |
| 2 | 1 (Na₃Cit) | 50 | 5 | $K_2CO_3$ | $H_2O$ | 97 |
| 3 | 0.1 (Na₃Cit) | 50 | 5 | $K_2CO_3$ | $H_2O$ | 65 |
| 4 | 0.05 (Na₃Cit) | 50 | 5 | $Cs_2CO_3$ | $H_2O$ | 89 |
| 5 | 0.05 (Na₃Cit) | 50 | 7 | $Cs_2CO_3$ | $H_2O$ | 98 |
| 6 | 0.05 (Na₃Cit) | 50 | 5 | CsOH | $H_2O$ | 66 |
| 7 | 0.05 (Na₃Cit) | 40 | 12 | $Cs_2CO_3$ | $H_2O$ | 48 |
| 8 | 0.05 (Na₃Cit) | 40 | 24 | $Cs_2CO_3$ | $H_2O$ | 94 |
| 9 | 0.1 (Na₃Cit) | 40 | 12 | $Cs_2CO_3$ | $H_2O$ | 76 |
| 10 | 0.05 (Na₃Cit) | 100 | 1 | $Cs_2CO_3$ | $H_2O$ | 80 |
| 11 | 0.05 (Na₃Cit) | 100 | 1.5 | $Cs_2CO_3$ | $H_2O$ | 98 |
| 12 | 0.05 (Na₃Cit) | 25 | 24 | $Cs_2CO_3$ | $H_2O$ | 13 |
| 13 | 0.05 (PEG) | 50 | 7 | $Cs_2CO_3$ | $H_2O$ | 89 |
| 14 | 0.05 (No) | 50 | 7 | $Cs_2CO_3$ | $H_2O$ | 57 |
| 15 | 0.05 (PVP) | 50 | 7 | $Cs_2CO_3$ | $H_2O$ | 37 |

[a] The reactions were conducted using 0.5 mmol of bromobenzene, 0.6 mmol of phenylboronic acid, and 3 mL of $H_2O$. [b] Analyzed using GC–MS.

Based on the results of 5-times recyclability test, the yield did not change significantly. This showed that $Na_3Cit$–$Fe_3O_4$/Pd has good effectiveness for the Suzuki coupling reaction (Figure 13).

**Figure 13.** Recycling test of $Na_3Cit$–$Fe_3O_4$/Pd catalyst.

## 6. Flower-Like Pd–$Fe_3O_4$ and Pd–$Fe_3O_4$ Hybrid Nanocatalyst-Embedded Au Nanoparticles

Several attempts have been taken to design new hybrid nanocomplexes with well-defined multicomponents by controlling the size and shapes of these materials through solution growth structure [89–91]. Nasrollahjadeh et al. [4] reported on the eco-friendly synthesis of Pd/$Fe_3O_4$ NPs using *Euphorbia condylocarapa M.* bieb root extract and applied as a magnetically recyclable catalyst for Suzuki and Sonogashira coupling reaction. Hoseini et al. [92] worked on the magnetic Pd/$Fe_3O_4$/r–GO nanocomposite as an effective and environmental catalyst for the Suzuki–Miyaura coupling reaction in water. Jang et al. [23] reported on facile synthesize Pd–$Fe_3O_4$ heterodimer as a magnetically recoverable catalyst for C–C coupling reaction. The use of hybrid Pd–$Fe_3O_4$ catalyst for C–C coupling reaction has also been vigorously researched through hyperbranched polyglycerol-inserted Pd–$Fe_3O_4$@$SiO_2$ [30]. Yeo et al. [93] developed a Pd–$Fe_3O_4$ core–satellite heterostructure as an effective candidate for the decarboxylative coupling reaction in aqueous solution. We focused on the synthesis of Pd–$Fe_3O_4$ by controlling shape and then the immobilization of Au NPs onto this Pd–$Fe_3O_4$ support

(Scheme 7). The synthesized Pd–Fe$_3$O$_4$ catalyst showed good catalytic performance for Sonogashira coupling reactions. On the other hand, the Au/Pd–Fe$_3$O$_4$ hybrid nanocomposites exhibited excellent catalytic performance for the tandem synthesis of 2-phenylindoles with great magnetic recoverability.

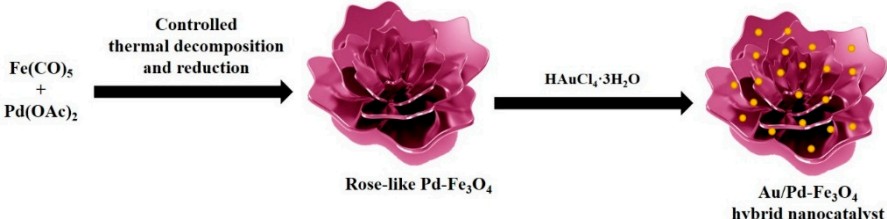

**Scheme 7.** Synthetic scheme of Pd–Fe$_3$O$_4$ and Au/Pd–Fe$_3$O$_4$ nanocatalyst.

### 6.1. Synthesis of Pd–Fe$_3$O$_4$ and Au/Pd–Fe$_3$O$_4$ Nanocomposites

Figure 14a,b depicts the scanning electron microscopy (SEM) images of the flower-like Pd–Fe$_3$O$_4$ and Au/Pd–Fe$_3$O$_4$ hybrid nanocomplexes. Au NPs were regularly distributed onto the Pd–Fe$_3$O$_4$ supports. Figures 13d and 14c exhibits the TEM images of Pd–Fe$_3$O$_4$ and Au/Pd–Fe$_3$O$_4$. The TEM image shows Pd–Fe$_3$O$_4$ nanocomplexes with a general Fe/Pd ratio of 64:36 (Figure 14c). The immobilization of Au NPs can be visualized from Figure 14d. Figure 14e displays the XRD patterns of the nanocomposites with total element ratios of 7:77:16 (Au/Fe/Pd) and 64:36 (Fe/Pd). The XRD pattern of Pd–Fe$_3$O$_4$ matched well with the Pd crystal structure and lattice planes of the cubic spinel structure of Fe$_3$O$_4$ (JCPDS no. 19-0629). On the other hand, the immobilized Au NPs are match well with fcc Au crystallizations (JCPDS no. 04-0784).

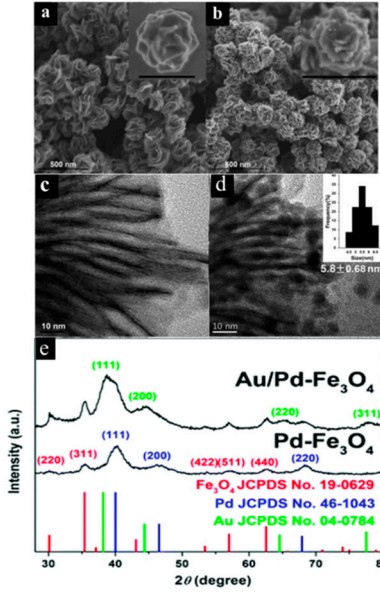

**Figure 14.** SEM images of flower-like Pd–Fe$_3$O$_4$ (**a**), Au/Pd–Fe$_3$O$_4$ (**b**) nanocomposites, TEM images of flower-like Pd–Fe$_3$O$_4$ (**c**), Au/Pd–Fe$_3$O$_4$ (**d**). Reproduced with permission from Park, *Solid State Sciences*, published by Elsevier, 2016.

### 6.2. Flower-Like Pd–Fe$_3$O$_4$: Application in Sonogashira Coupling Reactions

To assess the catalytic performance of the as-prepared Pd–Fe$_3$O$_4$ catalyst, the Sonogashira reactions of iodobenzene and phenylacetylenes were chosen as a model reaction under numerous conditions (Table 8). While optimizing the consequence of solvent, it was observed that the use of more polar solvent gave more yields, which is due to the great solubility of reactant and nanocatalyst in the reaction mixture (entries 1–3, Table 8). The influences of different bases were also studied

(entries 3–6, Table 8). The catalyst also showed good activity although when the temperature was reduced to 90 °C (entry 7, Table 8).

**Table 8.** Optimized reaction condition. Reproduced with permission from Park, *Solid State Sciences*, published by Elsevier, 2016.

| Entry | Cat. (mol %) | Temp (°C) | Time (h) | Base | Solvent | Conv. (%) [a] |
|-------|--------------|-----------|----------|------|---------|----------------|
| 1 | 1 | 120 | 18 | Piperidine | DMF | 72 |
| 2 | 1 | 120 | 18 | Piperidine | NMP | 76 |
| 3 | 1 | 120 | 18 | Piperidine | DMSO | 99 |
| 4 | 1 | 120 | 18 | $Cs_2CO_3$ | DMSO | 73 |
| 5 | 1 | 120 | 18 | NaOAc | DMSO | 98 |
| 6 | 1 | 120 | 18 | $K_2CO_3$ | DMSO | 78 |
| 7 | 1 | 90 | 18 | Piperidine | DMSO | 92 |
| 8 | 0.5 | 120 | 3 | Piperidine | DMSO | 99 |
| 9 | 0.25 | 120 | 3 | Piperidine | DMSO | 94 |
| 10 | 0.5 | 120 | 1 | Piperidine | DMSO | 93 |

Reaction conditions: iodobenzene (1mmol), phenylacetylene (1.1 mmol), base (2 mmol), solvent (5 mL). [a] Determined using gas chromatography–mass spectrometry (GC–MS).

Next, we tried to find the optimized conditions to check Turn over frequency (TOF). We then analyzed the impact of the catalyst amount and the reaction time. A decrease in the product conversion was observed when utilizing less catalyst and shorter reaction times (entries 9 and 10, Table 8). The Pd–$Fe_3O_4$ (TOF: 66.7) exhibited good catalytic efficiency than when Pd–$Fe_3O_4$ (TOF: 18) was synthesized by microbes and heterodimeric Pd/$Fe_3O_4$ (TOF: 4.2) [4,58].

The scope of substrate in Sonogashira coupling reaction catalyzed by Pd–$Fe_3O_4$ has been displayed in Table 9. Under our catalytic conditions, both electron donating and withdrawing substituents were smoothly coupled with arylacetylene with good conversion rates (entries 2–7, Table 9). Electron withdrawing substituents can normally produce advantageous consequences in Pd-catalyzed reactions by facilitating the rate-limiting oxidation step [94]. When the m-$CH_3$ and –$CF_3$ groups were utilized, product conversion was slightly improved but, comparatively, the conversion was good in the case of the m-$CF_3$ group (entries 8 and 9, Table 9).

**Table 9.** Sonogashira reaction of numerous aryl halides with arylacetylenes catalyzed by Pd–$Fe_3O_4$ nanocomposite. Reproduced with permission from Park, *Solid State Sciences*; published by Elsevier, 2016.

| Entry | Aryl Halide | Arylacetylene | Product | Conversion (%) [a] |
|-------|-------------|---------------|---------|---------------------|
| 1 |  |  |  | 72 |
| 2 |  |  |  | 76 |
| 3 |  |  |  | 99 |
| 4 |  |  |  | 73 |

**Table 9.** *Cont.*

| Entry | Aryl Halide | Arylacetylene | Product | Conversion (%) [a] |
|-------|-------------|---------------|---------|--------------------|
| 5 | H₃C─⬡─I | ⬡─≡ | H₃C─⬡─≡─⬡ | 98 |
| 6 | naphthalene | ⬡─≡ | naphthalene-≡-⬡ | 78 |
| 7 | H₃C─⬡─I | ─⬡─≡ | H₃C─⬡─≡─⬡ | 92 |
| 8 | F₃C─⬡─I | ⬡─≡ | H₃C...⬡─≡─⬡ | 99 |
| 9 | F₃C─⬡─I | ⬡─≡ | F₃C...⬡─≡─⬡ | 94 |

Reaction conditions: aryl halides (1.0 mmol), arylacetylene (1.1 mmol), piperidine (2.0 mmol) DMSO (5 mL), 120 °C, and 3 h. [a] Determined using gas chromatography–mass spectrometry (GC–MS).

## 6.3. Pd–Fe₃O₄ Supported Au Nanocatalyst: Applications for Tandem Synthesis of 2-Phenylindoles

To appraise the catalytic efficiency of the Au/Pd–Fe₃O₄ nanocatalyst, the tandem reaction of 2-phenylindoles with phenylacetylenes were exhibited as a model reaction under dissimilar environments (Table 10).

**Table 10.** Tandem synthesis of 2-phenylindoles. Reproduced with permission from Park, *Nanoscale*, published by The Royal Society of Chemistry, 2015.

| Entry | Catalyst | Temp (°C) | Time (h) | Base | Conversion (%) [a] |
|-------|----------|-----------|----------|------|--------------------|
| 1 | Pd–Fe₃O₄ | 120 | 18 | Piperidine | Trace [b] |
| 2 | Pd–Fe₃O₄ | 120 | 18 | Piperidine | 3 [c] |
| 3 | Pd–Fe₃O₄ | 120 | 18 | Piperidine | 41 |
| 4 | Pd–Fe₃O₄ | 120 | 18 | LiOAc | 45 |
| 5 | Pd–Fe₃O₄ | 120 | 18 | CsOAc | 48 |
| 6 | Au/Pd–Fe₃O₄ | 120 | 18 | CsOAc | 57 |
| 7 | Au/Pd–Fe₃O₄ | 150 | 18 | CsOAc | 97 |
| 8 | Au/Pd–Fe₃O₄ | 150 | 9 | CsOAc | 97 |
| 9 | Au/Pd–Fe₃O₄ | 150 | 6 | CsOAc | 59 |
| 10 | Au/Pd–Fe₃O₄ | 150 | 9 | CsOAc | 38 [d] |

Reaction conditions: Au/Pd–Fe₃O₄ catalyst (Au base: 0.18 mol %, Pd base: 0.5 mol %), 2-iodoaniline (0.5 mmol), phenylacetylene (0.6 mmol), base (1.0 mmol), DMSO (2.5 mL). [a] Determined using GC–MS spectroscopy based on 2-iodoaniline. [b] DMF was used as a solvent. [c] DMA was used as a solvent. [d] 0.09 mol % (Au base) of catalyst was used.

Compared to piperidine and LiOAc, CsOAc gave good conversion (48%). This can be explained from the hard–soft acid and base (HSCB) theory that e Cs⁺ is the best Pearson acid to eliminate iodide from the intermediate Pd NPs by maximizing the soft–soft interface [95]. The dual catalytic system exhibits higher catalytic performance than that of single catalytic system because of the electron transfer

across the interface. Hence, the Au/Pd–Fe$_3$O$_4$ catalyst exhibited good catalytic performance compared to Pd–Fe$_3$O$_4$ since Au NPs are very efficient in activating phenylacetylene (entries 5 and 6, Table 10) [96]. At high temperature (150 °C), the expected conversion (97%) was obtained (entry 7, Table 10). The effect of catalyst, as well as the reaction time, was also analyzed. A decrease of product conversion (59% and 38%) was observed while using 6 h and a lower amount of catalyst (entries 8–10, Table 10). The optimal reaction conditions were as follows: Au/Pd–Fe$_3$O$_4$ (Au base: 0.18 mol %; Pd base: 0.5 mol %); DMSO (2.5 mL), 150 °C, and 9 h (entry 8, Table 10). Table 11 exhibits the comparison of different Pd based catalyst in organic reaction.

**Table 11.** Comparison of different catalyst. Reproduced with permission from Park, *Nanoscale*, published by The Royal Society of Chemistry, 2015.

| SI No | Catalyst | Size (nm) | Morphology | Application |
|-------|----------|-----------|------------|-------------|
| 1 | Pd–Fe$_3$O$_4$ | 8.7 | Spherical | Mizoroki–Heck reaction |
| 2 | Pd–Fe$_3$O$_4$ | 213 | Flower-like | Sonogashira coupling reaction |
| 3 | Au/Pd–Fe$_3$O$_4$ | 5.8 | Flower-like | Tandem synthesis reaction |

## 7. Transition Metal Loading Pd–Fe$_3$O$_4$ Heterobimetallic Nanoparticles

Hybrid Pd/Fe$_3$O$_4$ nanoparticles (NPs) are the key factor in many catalytic reactions for organic transformation, due to the superior catalytic performance and magnetic recoverability [97]. Control of the composition, morphology, and architecture has attracted increasing attention in tailoring the resulting properties [98–100]. One of the developed methods is doped transition metal, and metal oxide on Pd–Fe$_3$O$_4$ for the synthesis of heterobimetallic NPs. Transition metal and metal oxide nanoparticles often provide an alternative to noble metals, with easy availability and low cost [101,102]. In addition, the composition of bimetallic and trimetallic NPs hybrid NPs compared to monometallic NPs is very promising and synchronously benefits from increasing selectivity, efficiency, and stability, owing to synergistic substrate activation [39,86,101,103–105]. NPs from transition metal loaded on hybrid Pd/Fe$_3$O$_4$ have been developed by our group and include Cu/Pd–Fe$_3$O$_4$ [38], Cu$_2$O/Pd–Fe$_3$O$_4$ [37], MnO/Pd–Fe$_3$O$_4$ [36], CoO/Pd–Fe$_3$O$_4$ [36], and Ni/Pd–Fe$_2$O$_3$ [39].

### 7.1. Synthesis of Hybrid Cu-Doped Pd–Fe$_3$O$_4$ Nanocatalyst

The synthesis of Cu-doped Pd–Fe$_3$O$_4$ nanocomposites were carried out by decomposition of Fe(CO)$_5$ and continued by the reduction of Pd(OAc)$_2$ and Cu(acac)$_2$ in presence of oleylamine (OAm) and 1-octadecene (ODE), and is outlined in Scheme 8a. The molar ratios of Pd/Fe/Cu were varied in the synthesis along with increasing amounts of NaOL. A high load of NaOL enlarged the BET surface areas of the Cu-doped Pd–Fe$_3$O$_4$ nanocomposites, demonstrating the result of NaOL on the surface area and morphology. Pd–Fe$_3$O$_4$ seed particles formed and aggregated, and nanosheets grew from the aggregate surface of Pd–Fe$_3$O$_4$. The morphology obtained without NaOL (Cu-doped Pd–Fe$_3$O$_4$-0) was spherical structures. In the presence of NaOL, Cu-doped Pd–Fe$_3$O$_4$ hybrid obtained a sheet-assembled formation, and not sphere-shaped (Scheme 8a). Cu-doped Pd–Fe$_3$O$_4$-*n* nanocomposites consist of crystal structures of fcc Pd crystal structure, and the cubic spinel structure of Fe$_3$O$_4$. The magnetic properties of the Cu-doped Pd–Fe$_3$O$_4$-0.3 nanocomposite have saturation magnetization value 9.2 emu g$^{-1}$. In addition, the remanence and coercivity of the hybrid nanocomplexes were close to zero, representative of superparamagnetism.

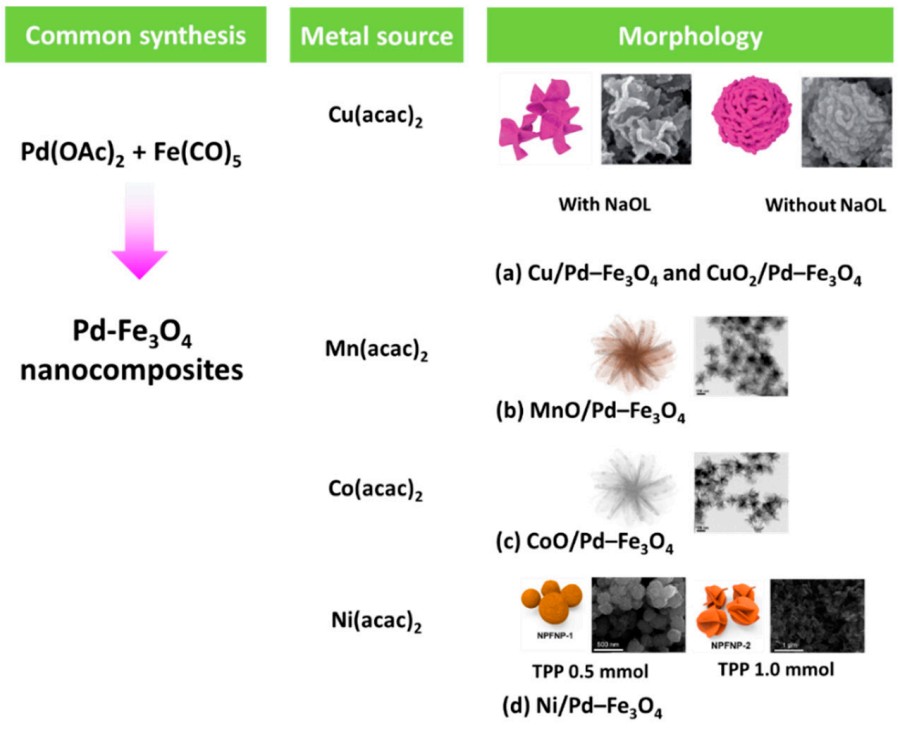

**Scheme 8.** Synthesis of transition metal-loaded Pd–Fe$_3$O$_4$ heterobimetallic nanoparticles.

## 7.2. Synthesize Cu$_2$O/Pd–Fe$_3$O$_4$ Nanocatalyst

The controlled thermal decomposition of iron pentacarbonyl and reduction of Pd and Cu (OAc)$_2$ were used in the synthesis of Cu$_2$O/Pd–Fe$_3$O$_4$ nanocomposites as shown in Scheme 8a. The Pd precursor and quantity of reducing agent affected the morphology of the nanocomposites. The flower-like morphology with a regular diameter of 173 nm was obtained using this method. Cu$_2$O/Pd–Fe$_3$O$_4$ nanocomposites confirm the fcc Pd crystal structure and cubic spinel structure of Fe$_3$O$_4$. Uniform distribution of Pd, Fe, and Cu substances over the whole nanocomposite verified the Cu$_2$O/Pd–Fe$_3$O$_4$ hybrid structure. The saturation magnetization analysis confirmed superparamagnetism. The specific surface areas of Cu$_2$O/Pd–Fe$_3$O$_4$ nanocomposites were higher compared to Pd–Fe$_3$O$_4$.

## 7.3. Hybrid MnO and CoO/Pd–Fe$_3$O$_4$ Nanocomplexes

The fabrication of MnO/Pd–Fe$_3$O$_4$ and CoO/Pd–Fe$_3$O$_4$ nanocomplexes were similar to previous work with the modification of metal precursors such as Mn(acac)$_2$ or Co(acac)$_2$ in OAm and ODE in Scheme 8b,c. The morphology of hybrid MnO/Pd–Fe$_3$O$_4$ and CoO/Pd–Fe$_3$O$_4$ obtained using this method corresponded to uniform hierarchical, and the nanosheets emitted small seed particles in the center (Scheme 8b,c).

The crystal structure of the nanocomposite established the fcc structure of Pd and the cubic spinel structure of Fe$_3$O$_4$ (Figure 15). The addition of metal source decreased the intensity of the Fe$_3$O$_4$ peak, attributing the crystallization of Fe$_3$O$_4$ to disorder caused by Mn and Co ions. The crystalline MnO and CoO did not appear, suggesting that the overhead oxides had an amorphous structure.

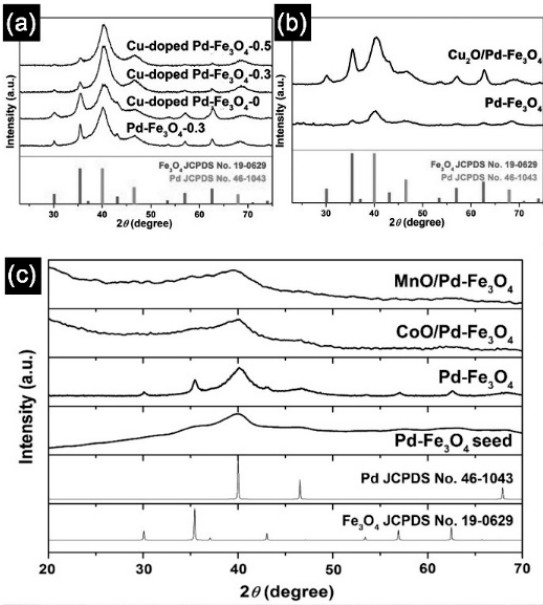

**Figure 15.** XRD spectrum of (**a**) Cu-doped Pd–Fe$_3$O$_4$, (**b**) Cu$_2$O-doped Pd–Fe$_3$O$_4$, and MnO-, CoO-doped Pd–Fe$_3$O$_4$, respectively. Reproduced with permission from Park, published by (**a**) *Journal of Materials Chemistry A*, The Royal Society of Chemistry, 2015; (**b**) *RSC Advances*, The Royal Society of Chemistry, 2016; and (**c**) *Catalysis Communications*, Elsevier, 2017.

### 7.4. Synthesis of Hybrid Ni–Pd–Fe$_3$O$_4$ Nanocomposites

Pd–Fe$_3$O$_4$ hybrids loaded with transition metal Ni were also fabricated using thermal decomposition and reduction methods. In this method, the amount of triphenylphosphine (TPP) had an effect on controlling the morphology of the nanocomposites. The amount of TPP used was 0.5 and 1 mmol. To synthesize Ni-doped Pd–Fe$_3$O$_4$ hybrid nanoparticles (NPFNPs), Pd(OAc)$_2$, Fe(CO)$_5$, and Ni(acac)$_2$ were used as a salt (Scheme 8d). Spherical morphology with a rough surface was obtained when 0.5 mmol TPP was used, and the average particle diameter around 244 ± 38 nm. Increasing TPP to 1 mmol resulted in the morphology changing to a more impressive nanosheet at the corners, which an average particle diameter of 215 ± 17 nm. On the other hand, the crumpled ball morphology regularly collapsed when applying an excess amount of TPP, confirming the amount of TPP plays an essential role in the morphology-controlled synthesis of nanocomposites.

### 7.5. Applications of Transition Metal-Loaded Pd–Fe$_3$O$_4$ Heterobimetallic Nanoparticles in Organic Reactions

Pd catalysis in organic transformations such as the Suzuki–Miyaura, Heck, Sonogashira, tandem reactions, hydroboration, etc., are essential in different organic synthesis procedures with huge interest and in many fields of application. As previously mentioned, the advantages of the combination of transition metal and metal oxide loading on Pd–Fe$_3$O$_4$ heterobimetallic nanoparticles, as see in Scheme 9, will be discussed.

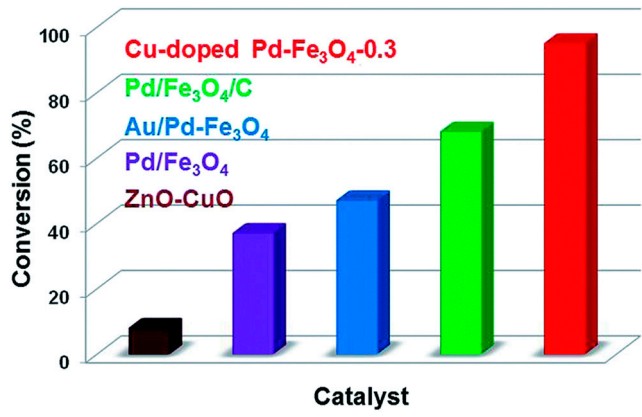

**Scheme 9.** Schematic applications of transition metal loading Pd–Fe$_3$O$_4$ heterometallic nanoparticles in organic reactions. Reproduced with permission from Park, published by (**a**) *Journal of Materials Chemistry A*, The Royal Society of Chemistry, 2015; (**b**) *RSC Advances*, The Royal Society of Chemistry, 2016; (**c**) Catalysis Communications, Elsevier, 2017; and (**d**) Catalysts, MDPI, 2017.

### 7.5.1. Tandem Synthesis of 2-Phenylbenzofurans

The catalytic performance of Cu-doped Pd–Fe$_3$O$_4$ was studied for tandem synthesis of 2-phenylbenzofurans from 2-iodophenols with phenylpropiolic acids as a model reaction in Scheme 8a. Regarding the Cu-doped Pd–Fe$_3$O$_4$-0.3 nanocomposite, the Cu-doped Pd provides impressive catalytic performance, and superior stability was achieved using recovery and leaching experiments. The comparison of the catalytic activity and conversion with heterogeneous catalysts used in earlier work by our group is shown in Figure 16.

**Figure 16.** Comparison of Cu-doped Pd–Fe$_3$O$_4$ catalytic activity with heterogeneous catalysts as previously reported by our group. Reproduced with permission from Park, *Journal of Chemistry A*, The Royal Society of Chemistry, 2015.

### 7.5.2. C–H Arylation of 1-Butyl-4-Nitro-1*H*-Imidazoles

The catalytic performance of Cu$_2$O/Pd–Fe$_3$O$_4$ was investigated for C–H arylation, where the benchmark substrates are 1-butyl-4-nitro-1*H*-imidazoles with iodobenzenes (Scheme 9b). Good catalytic performance

was observed for the hybrid Cu$_2$O/Pd–Fe$_3$O$_4$ catalyst compared with Pd–Fe$_3$O$_4$ or other catalysts, as shown in Table 12. The remarkable result is due to the electron transfer across the metal–oxide interface and the synergetic effect of Cu and Pd–Fe$_3$O$_4$.

**Table 12.** Oxidation of benzyl alcohol using various catalysts based on Cu$_3$(BTC)$_2$. Reproduced with permission from Park, *RSC Advances*, published by The Royal Society of Chemistry, 2016.

| Entry | Catalysts | Time (h) | Temp (°C) | Conv [a] (%) |
|-------|-----------|----------|-----------|--------------|
| 1 | Pd–Fe$_3$O$_4$ | 18 | 130 | 62 |
| 2 | Cu$_2$O/Pd–Fe$_3$O$_4$ | 18 | 130 | 73 |
| 3 | Cu$_2$O/Pd–Fe$_3$O$_4$ | 18 | 140 | 85 [b] |
| 4 | Cu$_2$O/Pd–Fe$_3$O$_4$ | 9 | 140 | 84 [b] |
| 5 | Cu$_2$O/Pd–Fe$_3$O$_4$ | 4.5 | 140 | 76 [b] |
| 6 | Cu$_2$O/Pd–Fe$_3$O$_4$ | 9 | 140 | 76 [b, c] |
| 7 | Pd/charcoal | 4.5 | 140 | 69 [b] |
| 8 | Fe$_3$O$_4$/charcoal | 4.5 | 140 | 0 [b] |
| 9 | Cu$_2$O | 4.5 | 140 | 0 [b] |

Reaction conditions: catalyst (Pd base: 5.0 mol %), 1-butyl-4-nitro-1*H*-imidazole (0.5 mmol), iodobenzene (0.55 mmol), base NaOAc (1.0 mmol), solvent DMSO (3.0 mL). [a] Determined by $^1$H NMR. [b] Iodobenzene (1.0 mmol), NaOAc (2.0 mmol) and DMSO (5 mL) were used. [c] 2.5 mol % of catalyst was used.

The Cu$_2$O/Pd–Fe$_3$O$_4$ catalyst also showed good conversion with numerous substituted aryl iodides, as shown in Table 13. The effect of electron-donating substituents and electron-deficient progressed with high reactivity, except the COMe group.

**Table 13.** Substrate scope. Reproduced with permission from Park, *RSC Advances*, published by The Royal Society of Chemistry, 2016.

| Entry | Substrate | Conv. (%) | Entry | Substrate | Conv. (%) |
|-------|-----------|-----------|-------|-----------|-----------|
| 1 | | 84 | 6 | | 70 |
| 2 | | 79 | 7 | | 39 |
| 3 | | 84 | 8 | | 71 |

**Table 13.** *Cont.*

| Entry | Substrate | Conv. (%) | Entry | Substrate | Conv. (%) |
|-------|-----------|-----------|-------|-----------|-----------|
| 4 | | 85 | 9 | | 80 |
| 5 | | 75 | 10 | | 69 |

Reaction conditions: catalyst (Pd base: 5.0 mol %), 1-butyl-4-nitro-1*H*-imidazole (0.5 mmol), iodobenzene (1.0 mmol), NaOAc (2.0 mmol), DMSO (5.0 mL).

### 7.5.3. Synthesis of Alkylboronates from Styrene

The hydroboration of styrene with $B_2Pin_2$ was used in the catalytic test of $MnO/Pd–Fe_3O_4$ and $CoO/Pd–Fe_3O_4$ (Table 14). THF is efficiently catalyzed this reaction. MeOH acts as a hydrogen donor. The high yield was obtained while using as an additive (entry 2) and not as a solvent (entry 1). The best base was $Cs_2CO_3$ with the highest yield of 67%. Furthermore, the hybrid $CoO/Pd–Fe_3O_4$ was observed to have better catalytic performance than the other nanocomposites $MnO/Pd–Fe_3O_4$, $Pd–Fe_3O_4$, and Pd/charcoal, due to electron transfer beyond the metal and oxide interface. The recyclability and leaching tests suggest the superior catalytic performance and impressive stability of the nanocatalyst.

**Table 14.** Synthesis of alkylboronates from styrene. Reproduced with permission from Park, *Catalysis Communications*, published by Elsevier, 2017.

| Entry | Catalysts | Solvent | Base | Time (h) | Yield (%) [a] |
|-------|-----------|---------|------|----------|-----------|
| 1 | $CoO/Pd–Fe_3O_4$ | MeOH | KOtBu | 12 | Trace |
| 2 | $CoO/Pd–Fe_3O_4$ | THF | KOtBu | 12 | 33 |
| 3 | $CoO/Pd–Fe_3O_4$ | THF | NaOMe | 12 | 36 |
| 4 | $CoO/Pd–Fe_3O_4$ | THF | $Cs_2CO_3$ | 12 | 67 |
| 5 | $MnO/Pd–Fe_3O_4$ | THF | $Cs_2CO_3$ | 12 | 29 |
| 6 | $Pd–Fe_3O_4$ | THF | $Cs_2CO_3$ | 12 | 35 |
| 7 | Pd/charcoal | THF | $Cs_2CO_3$ | 12 | Trace |
| 8 | $CoO/Pd–Fe_3O_4$ | THF | $Cs_2CO_3$ | 24 | 63 |

Reaction conditions: styrene (1.0 mmol), $B_2Pin_2$ (1.1 mmol), catalyst (Pd base: 1.0 mol %), base (2.0 mmol), MeOH (5 mmol), solvent (3.0 mL) 60 °C, 12 h. [a] Isolated yields.

### 7.5.4. Suzuki–Miyaura Coupling Reaction

The catalytic properties of $Ni/Pd–Fe_3O_4$ applied in Suzuki–Miyaura C–C coupling reaction were studied using bromobenzene and phenylboronic acid. A combination of water and water (1:1) at 50 °C (Scheme 9d) was used. The yields of the products and TOFs for the reactions containing the NPFNP-1 Ni-doped $Pd–Fe_3O_4$ hybrid nanoparticles (NPFNPs), NPFNP-2, and $Pd–Fe_3O_4$ as the catalyst was obtained under giving conditions. NPFNP-2 presented relatively high catalytic performance compared to NPFNP-1 and $Pd–Fe_3O_4$ with a similar mol % of Pd as shown in Figures 16b and 17a. The high catalytic activity of NPFNP-2 can be indicative of a higher number of surface deficiencies caused by the changes in morphology and the synergistic properties of individual components. The defect surface and synergistic effects facilitated the oxidative addition reaction of aryl halide resulting in an increase in yield.

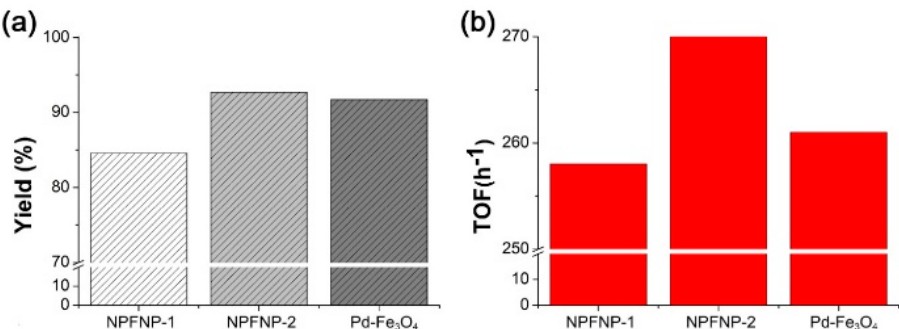

**Figure 17.** Comparison of (**a**) yield and (**b**) TOF for the products in the Suzuki–Miyaura coupling reaction. Reproduced with permission from Park, *Catalysts*, published by MDPI, 2017.

## 8. Conclusions

The synthesis of hybrid $Pd–Fe_3O_4$ nanoparticles was reviewed with a focus on urchin-like $FePd–Fe_3O_4$, $Pd/Fe_3O_4$, $Pd/Fe_3O_4$/charcoal, flower-like $Pd–Fe_3O_4$, and transition metal-loaded $Pd–Fe_3O_4$ nanocomposites which act as successful catalysts for various C–C coupling reactions. We reviewed all of the hybrid $Pd–Fe_3O_4$ nanocomposites that showed better catalytic performance and reusability than many previously reported catalysts, because of magnetic properties. Hybrid $Pd–Fe_3O_4$ NPs exhibited high performance, stability, and recyclability with respect to morphology and magnetic properties. We anticipate that our methodology can be developed for other catalytic systems in the near future.

**Author Contributions:** K.H.P., J.C.P. and S.P. provided academic direction and worked hardly to get the fundings. S.J. and S.A. contributed equally on that work. Specially, S.J. collected materials and wrote the introduction and conclusion part. S.A. contributed in results and discussion part and checked English style as well as grammatical errors throughout the manuscript. D.A., S.S. and M.Y. contributed to the materials and methods and results and discussion part.

**Funding:** This research was financially supported by Basic Science Research Program as well as National Research Foundation of Korea (NRF). It was also supported by the Ministry of Science, ICT & Future Planning (NRF-2017R1A4A1015533 and NRF- 2017R1D1A1B03036303 and NRF-2018R1D1A1B07045663). This study was supported in part by NRF Korea (NRF-2018R1D1A1B07045663) and Korea Basic Science Institute (KBSI) grant (C38529) to Sungkyun Park.

**Conflicts of Interest:** The authors declare no conflict of interest.

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
