# Peer review of "Recent Novel Hybrid Pd–Fe3O4 Nanoparticles as Catalysts for Various C–C Coupling Reactions"

_processes, doi:10.3390/pr7070422_

Round 1

Reviewer 1 Report

The authors propose a review based abundantly on their very recent work, to the limit of reporting basically all the figures and tables of some references. I'm not very comfortable with this. I'd suggest to report more briefly their previous findings and include recent works on the topic by other authors. English language needs check.

Author Response

Thanks for your efficient suggestions. Here, mostly we tried to work on the papers which are published from our paper. Besides, we have added some papers from other authors. Anyway, on the basis of your suggestions we are adding some recent paper also.

Reviewer 2 Report

The paper «A Recent novel hybrid Pd-Fe3O4 nanoparticles as catalysts for various C-C coupling reaction» written by Jang S., Hira S.A., Annas D., Song S., Yusuf M., Park S. and Park K.H. presents data about new hybrid Pd-Fe3O4 nanoparticles obtaining. The justification of multimetallic complexes involved usage as high-effective C-C coupling reactions catalysts, comparison of such complexes with well-known analogous particles and analysis of this compounds usage advantages such as presence of both high saturation magnetization and magnetic connectivity, heterogeneous reaction course, milder conditions for catalytic reactions, ability to surface morphology controlling are presented there. Detail described the method of composite nanoparticles and catalysts based on them preparation and study of their magnetic and catalytic properties also described in this manuscript. The authors provide numerous examples of obtaining catalysts synthesized in various C-C coupling reactions. By varying such conditions as temperature, solvent, reaction time there’s become possible to obtain target products with the highest yields, selecting the optimal conditions for catalyst usage. Much attention is paid to catalysts properties investigation. Multimetallic complexes described shown to exhibit high stability and efficiency as well as greater recyclability related to specific morphological and magnetic properties. Carrying out experimental part, authors rely on such reliable research methods as scanning electron microscopy, X-ray diffraction, magnetic hysteresis analysis.  The results and conclusions obtained are in a good agreement with fundamental laws and concepts of chemistry and clearly articulated and confirmed. Finally, specific practical significance of hybrid nanoparticles synthesized is described in this paper; the assumption of their near future usage as high-effective coupling reactions catalysts is promoted there.

The manuscript is written at proper level in competent English and fully complies with the subject matter of journal Processes and its requirements. From the notes it should be noted the authors should systematize the alignment of punctuation marks and tabs, bring everything to a uniformity. The article is recommended for publication with minor revision.

Author Response

Thanks for your nice complements. According to your suggestions, the alignment of punctuation marks and tabs has been brought uniformly. Besides, the manuscript has been checked properly with more care.

Round 2

Reviewer 1 Report

The authors have partially addresses my previuos concerns. I think the manuscript in the current version can be published after minor english editing.